# Molecular Evolutionary Analyses of the *Pseudomonas*-Derived Cephalosporinase Gene

**DOI:** 10.3390/microorganisms11030635

**Published:** 2023-03-01

**Authors:** Tatsuya Shirai, Mao Akagawa, Miho Makino, Manami Ishii, Ayaka Arai, Norika Nagasawa, Mitsuru Sada, Ryusuke Kimura, Kaori Okayama, Taisei Ishioka, Haruyuki Ishii, Shinichiro Hirai, Akihide Ryo, Haruyoshi Tomita, Hirokazu Kimura

**Affiliations:** 1Advanced Medical Science Research Center, Gunma Paz University Research Institute, Shibukawa 377-0008, Gunma, Japan; 2Department of Respiratory Medicine, Kyorin University School of Medicine, Mitaka 181-8611, Tokyo, Japan; 3Department of Health Science, Gunma Paz University Graduate School of Health Sciences, Takasaki 370-0006, Gunma, Japan; 4Department of Medical Technology, Gunma Paz University School of Medical Science and Technology, Takasaki 370-0006, Gunma, Japan; 5Department of Bacteriology, Gunma University Graduate School of Medicine, Maebashi 371-8514, Gunma, Japan; 6Department of Agriculture, Takasaki University of Health Welfare, Takasaki 370-0033, Gunma, Japan; 7Infectious Disease Surveillance Center, National Institute of Infectious Diseases, Musashimurayama 162-8640, Tokyo, Japan; 8Department of Microbiology, Yokohama City University School of Medicine, Yokohama 236-0004, Kanagawa, Japan

**Keywords:** *Pseudomonas aeruginosa*, PDC, evolution

## Abstract

Despite the increasing evidence of the clinical impact of *Pseudomonas*-derived cephalosporinase (PDC) sequence polymorphisms, the molecular evolution of its encoding gene, *bla*_PDC_, remains elusive. To elucidate this, we performed a comprehensive evolutionary analysis of *bla*_PDC_. A Bayesian Markov Chain Monte Carlo phylogenetic tree revealed that a common ancestor of *bla*_PDC_ diverged approximately 4660 years ago, leading to the formation of eight clonal variants (clusters A–H). The phylogenetic distances within clusters A to G were short, whereas those within cluster H were relatively long. Two positive selection sites and many negative selection sites were estimated. Two PDC active sites overlapped with negative selection sites. In docking simulation models based on samples selected from clusters A and H, piperacillin was bound to the serine and the threonine residues of the PDC active sites, with the same binding mode for both models. These results suggest that, in *P. aeruginosa*, *bla*_PDC_ is highly conserved, and PDC exhibits similar antibiotic resistance functionality regardless of its genotype.

## 1. Introduction

*Pseudomonas aeruginosa* (Pseudomonadaceae) is a major opportunistic pathogen causing severe nosocomial infections, such as ventilator-associated pneumonia, urinary tract infections, and bacteremia [1,2,3]. Treating these infections is challenging because *P. aeruginosa* exhibits innate resistance to a wide range of antibacterial agents [4] based on its outer membrane impermeability, efflux pump expression, and resistance-conferring enzyme production [5,6]. Its enzyme class C β-lactamase (*Pseudomonas*-derived cephalosporinase, PDC), encoded by *bla*_PDC_, confers resistance primarily to β-lactam antibiotics [7]. Moreover, in *bla*_PDC_, mutations in the peptidoglycan-recycling process cause hyperproduction of this enzyme, resulting in reduced susceptibility to antibiotics against *P. aeruginosa*, including ceftazidime, cefepime, and piperacillin [8,9,10,11].

PDC contains more than 500 variants according to the Beta-Lactamase DataBase (http://bldb.eu/BLDB.php?prot=C#PDC, accessed on 26 December 2022) that can lead to its structural modifications [12,13]. Moreover, the PDC sequence polymorphism can alter resistance to antibiotics, thereby resulting in a major obstacle in *P. aeruginosa* treatment in clinical settings [12,14,15,16]. However, the evolutionary history of *bla*_PDC_ and the molecular interactions between PDC and the relevant antibiotics for each *bla*_PDC_ genotype remain elusive.

Recent advances in in silico techniques and computing have resulted in impressive progress in molecular-evolutionary and docking-simulation analyses. These analyses provide opportunities for deciphering the evolution of PDCs and their molecular mechanisms of antibiotic resistance. We, therefore, analyzed the molecular evolution of *bla*_PDC_ collected from different parts of the world. For the *bla*_PDC_ genotypes selected from specific clonal variants (clusters), we clarified its molecular interactions with piperacillin, a major antibiotic against *P. aeruginosa*, via a docking simulation between PDC and piperacillin.

## 2. Materials and Methods

### 2.1. Sample Selection and Alignment

To investigate the molecular evolution of *bla*_PDC_, we obtained all available full-length nucleotide sequences for it from GenBank (https://www.ncbi.nlm.nih.gov/genbank/, accessed on 19 May 2021). Samples with missing data regarding the years or regions of isolation or detection or exhibiting ambiguous sequences or genetic recombination were omitted, leaving 511 samples. MEGA7 [17] was used to generate amino acid sequences for these samples, and multiple alignment was conducted using MAFFT 7 [18]. The genome data with homologous sequences (100% identity) were excluded using Clustal Omega [19], and the genome that was first isolated was also left. In total, 215 samples, isolated or detected in 40 countries between 1963 and 2021, were used. The present genome sample information is shown in Appendix A.

### 2.2. Time-Scaled Phylogenetic Tree Analysis

A time-scaled phylogenetic tree was constructed using the Bayesian Markov Chain Monte Carlo (MCMC) method in BEAST 2.4.8 [20], with a chain length of 40,000,000 steps and sampling every 1000 steps. First, using jModelTest 2.1.10 [21], we selected the best substitution model (TPM3uf + I + G). Subsequently, we applied the path-sampling/stepping-stone sampling method [22,23] to search for the optimal combination of four clock models (Strict Clock, Relaxed Clock Exponential, Relaxed Clock Log Normal, and Random Local Clock) and two prior tree models (Coalescent Constant Population and Coalescent Exponential Population). For the analyses, we selected the Strict Clock and Coalescent Exponential Populations models. We then used Tracer 1.7.1 [24] to assess the effective sample size with respect to convergence, accepting values >200. After discarding the first 15% of iterations as burn-in, phylogenetic trees were generated using Tree Annotator 2.4.8 implemented in BEAST. Subsequently, the trees were illustrated using FigTree 1.40 (http://tree.bio.ed.ac.uk/software/figtree/, accessed on 12 October 2021). Moreover, we selected the representative *bla*_PDC_ samples from each cluster determined by MCMC tree (GenBank: Cluster A, NZ_JAHBSB010000003; cluster B, NZ_RRBX01000028; cluster C, NZ_PHTA01000011; cluster D, NZ_CP053917; cluster E, NG_055276; cluster F, NZ_JAGFEQ010000011; cluster G, NZ_JAGMWR010000017; cluster H, CP020560). Then, their PDC sequences were compared. The evolutionary rate of the selected *bla*_PDC_ samples was calculated using Tracer 1.7.1 by selecting a suitable model, as described.

### 2.3. Phylogenetic Distance Analysis

To calculate the phylogenetic distances among the samples, a phylogenetic tree was created using the maximum likelihood (ML) method in MEGA7. First, the best substitution model was selected using jModelTest 2.1.10. The phylogenetic distances of the ML tree, and for each cluster, were estimated using Patristic [25].

### 2.4. Structure Retrieval and Modeling

We obtained the three-dimensional (3D) structures of PDC (PDC-1, PDBID:4WYY) as a template for homology modeling from the Protein Data Bank Japan (https://pdbj.org/, accessed on 8 December 2021). The 3D structure of piperacillin was downloaded from PubChem (https://pubchem.ncbi.nlm.nih.gov/, accessed on 9 December 2021). The *bla*_PDC_ samples used for structural modeling were from clusters separated by large genetic distances (based on the MCMC tree and phylogenetic distance), namely clusters A and H. Structural models of PDC encoded by these two *bla*_PDC_ were constructed using the template model using MODELLER 10.2 [26]. The structural reliability of the constructed models was evaluated using Ramachandran plot analysis in CooT 0.8.9.2 [27]. Subsequently, energy minimization was performed using GROMOS96 implemented in Swiss PDB Viewer 4.1.0 [28]. Based on prior reports [29,30,31,32], three active sites of PDC were identified as follows: Ser62, Val63, Ser64, and Lys65 (here, motif 1), Tyr149, Ser150, and Asn151 (here, motif 2), Lys314, Thr315, and Gly316 (here, motif 3).

### 2.5. Selective Pressure Analyses

To reveal the positive and negative selection sites of PDC amino acid sequences, non-synonymous (dN) and synonymous (dS) substitution rates were estimated using the Datamonkey web server (http://www.datamonkey.org/, accessed on 6 December 2021). We used single-likelihood ancestor counting (SLAC), fixed-effects likelihood (FEL), internal fixed-effects likelihood (IFEL), fast unconstrained Bayesian approximation (FUBAR), and mixed-effects model of evolution (MEME) to estimate positive selection sites, and SLAC, FEL, IFEL, and FUBAR to estimate negative selection sites. The determination of positive (dN/dS > 1) and negative (dN/dS < 1) selection was based on *p* < 0.05 for SLAC, FEL, IFEL, and MEME, and posterior probabilities > 0.9 for FUBAR. These selection sites were mapped onto the structural models of PDCs.

### 2.6. Docking Simulation

To elucidate the molecular interactions between the structural models of PDC and piperacillin, docking simulations were performed using AutoDock Vina 1.1.2 [33]. Prior to docking simulation, polar hydrogen atoms and Gasteiger charges were added to PDC models using AutoDockTools 1.5.6. The grid box size was set to include all proteins. After the 20 docking models were generated using AutoDock Vina, 3D molecular interactions and conformations were visualized using PyMOL 2.3.4. Docking models with root mean square deviation ≥2, relative to the values obtained prior to docking simulation, were omitted. The optimal models in each cluster were determined from the docking models using AutoDock Vina based on the binding energy. The molecular interactions were analyzed in a 2D diagram using the BIOVIA Discovery Studio Visualiser.

## 3. Results

### 3.1. Time-Scaled Phylogenetic and Evolutionary Analysis

The MCMC time-scaled phylogenetic tree of *bla*_PDC_ (Figure 1) separated *P. aeruginosa* into eight clusters (A to H), with a common ancestor diverging 4661 years ago [95% highest posterior density (HPD), 2743–6675 years ago], after which cluster H emerged. The other clusters emerged as follows (in years ago, with 95% HPD interval): Cluster G, 537 (332–776); F, 437 (287–649); E, 355 (238–528); D, 192 (134–296); C, 192 (134–296); B, 173 (128–268); A 173 (128–268). The *bla*_PDC_ belonging to cluster A were predominant at the time of analysis. Moreover, the active sites (motifs 1 to 3) of representative PDC sequences from each cluster were found to be conserved (Appendix A). The evolutionary rate (in substitutions/site/year) of the entire *bla*_PDC_ gene was 4.08 × 10^−5^ (95% HPD, 2.52 × 10^−5^ to 5.68 × 10^−5^).

### 3.2. Time-Scaled Phylogenetic and Evolutionary Analysis

The phylogenetic distance among all of the *bla*_PDC_, estimated using an ML-based phylogenetic tree (Figure 2a), was 0.013 ± 0.013 (mean ± SD). For clusters A to G (Figure 2b), the mean distances were short, ranging from 0.005 to 0.086, with cluster A having the shortest phylogenetic distance (0.005 ± 0.000) and H having a relatively long distance (0.44 ± 0.046).

### 3.3. Selective Pressure Analysis

We estimated the positive and negative selection sites for the present PDC using the Datamonkey web server. Two amino acid substitutions (amino acids 79 and 239), common to all five methods, were identified as positive selection sites. The number of negative selection sites was 18, 54, 48, and 56 for SLAC, FEL, IFEL, and FUBAR, respectively. Of these, 18 sites were common to all four methods. No positive or negative selection sites were located in motif 1. In contrast, negative selection sites Tyr149, Thr315, and Gly316 were located in motif 2 and motif 3, respectively (Figure 3).

### 3.4. Molecular Interactions between PDC and Piperacillin

Docking simulation between the PDC structural models and piperacillin to clarify their molecular interactions was performed using AutoDock Vina. Based on the phylogenetic analysis results, PDC structural models were generated from the representative samples selected from clusters A and H.

In the PDC structural models of cluster A, piperacillin interacted with Ser62 (motif 1) and Thr315 (motif 3) at the active sites via conventional hydrogen bonds (Figure 4a). An unfavorable interaction (donor–donor) was formed with Asn345 in PDC. The forces that interacted with sites other than the active sites were carbon–hydrogen bonds and hydrophobic interactions (π–π stacked and alkyl). The interacting residues other than those in the active sites were alanine, asparagine, serine, and tyrosine. The computational binding affinity was estimated to be −9.0 kcal/mol.

In the PDC structural models of cluster H, piperacillin formed conventional hydrogen bonds with Ser62 (motif 1) and Thr315 (motif 3) at the active sites (Figure 4b). There was an unfavorable interaction (donor–donor) between Arg348 in PDC and piperacillin. Carbon–hydrogen bonds and hydrophobic interactions (π–π stacked and alkyl) were formed as attractive forces at sites other than the active sites. The interacting residues other than those in the active sites were alanine, arginine, asparagine, serine, and tyrosine. The binding affinity was computed as −9.0 kcal/mol.

## 4. Discussion

Sequence polymorphism of PDC has generated increasing concern [34,35]. To elucidate the molecular evolution of *bla*_PDC_, we investigated this using *bla*_PDC_ gene samples from various parts of the world. Based on the *bla*_PDC_ gene clusters revealed by this analysis, we conducted a docking simulation between PDC and piperacillin to elucidate their molecular interactions. This revealed that PDC evolved constantly and formed eight clusters, whereas the phylogenetic distances of *bla*_PDC_ were short. These results, suggesting low levels of genetic divergence among the current *bla*_PDC_, are consistent with a prior study of clinically isolated samples, in which the sequences were relatively conserved despite variations in gene size [36]. At the same time, the number of PDC with altered resistance to antibiotics is increasing [34,35,37]. These findings indicate that even minor amino acid substitutions may affect the functions of PDC.

Our selection-pressure analysis of PDC revealed two positive selection sites and many negative selection sites. Moreover, negative selection sites overlapped with motifs 2 (Tyr149) and 3 (Thr315 and Gly316) in PDC active sites. Negative selection acts as a sieve, which leads to the elimination of harmful mutations [38,39,40]. This suggests that *P. aeruginosa* with mutations in PDC active sites do not adapt to their environment, leading to the presence of highly conserved active sites. Indeed, the active sites of representative PDC sequences from each cluster were found to be conserved.

Molecular evolutionary analysis of *P. aeruginosa gyrA* revealed that key residue substitutions in GyrA, with positive selection due to quinolone drugs, altered the binding mode between GyrA and ciprofloxacin [41]. Although PDC is an antibiotic resistance-conferring enzyme rather than a target, its molecular mechanisms of action may evolve under selective pressure. In our PDC–piperacillin docking simulation, piperacillin interacted with serine and threonine residues at the active sites for clusters A and H. These results suggest that PDC shows similar antibiotic resistance mechanisms regardless of its genotype. This is partially due to the conservation of active sites, which play an essential role in β-lactam antibiotic hydrolysis, among the clusters [42]. Furthermore, deeper insight into the molecular mechanisms of PDC in evolutionarily selected variants will pave the way towards developing promising β-lactam/β-lactamase inhibitors against *P. aeruginosa*.

This study has computing capability-related limitations that need to be addressed in the future. The first is the reliability of structural models. The homology modeling generates a good stereochemical protein model but is limited in predicting the native structure accurately from its sequence and template structure alone [43]. Quite recently, a new powerful tool of structural modeling using deep learning techniques, Alphafold2, has become available [44]. AlphaFold2 may exhibit a smaller structural gap between native and target proteins than homology modeling [44]. However, we could not perform AlphaFold2 because it requires high computing capability. The second limitation concerns the lack of a dynamic process through which PDC recognizes piperacillin in our simulation. Subtle changes in proteins may cause drug resistance by maintaining substrate recognition and catalysis while preventing the antibiotic from exerting its full effect [15,45]. In PDC, various amino acid substitutions at sites other than the active sites, such as those in the short peptide strand traversing the active sites (Ω loop) and the C-terminal domain, can affect resistance to β-lactam antibiotics [29,35]. We were unable to include these catalytic residues in our docking simulation because our method applies a “snap-shot” approach that does not present the molecular mechanisms of target molecules as a dynamic image. Although molecular dynamic simulations using extremely high-performance computer systems provide opportunities to explore dynamic molecular mechanisms [46,47,48], relatively few laboratories have the facilities to conduct them. Although the methodology of the present study is subject to these limitations, our molecular evolution and docking simulation analyses will guide the designing of effective therapies targeting PDC.

## 5. Conclusions

To the best of our knowledge, this is the first comprehensive evolutionary analysis of PDC gene. These findings will improve our understanding of time-scaled genetic and functional changes in PDC. Based on the MCMC phylogenetic tree, a common ancestor of *bla*_PDC_ diverged ca. 4660 years ago, giving rise to eight clusters. The phylogenetic distances within the samples were short. Our analysis revealed two positive selection sites, and many negative selection sites, in PDC. These were not located in the active sites motif 1 but in motifs 2 (Tyr149) and 3 (Thr315 and Gly316). Docking simulations between piperacillin and the PDC structural models based on clusters A and H revealed similar binding modes. These findings imply that *bla*_PDC_ is highly conserved, particularly at the active sites, resulting in similar antibiotic-related functions for PDC, regardless of the genotype.

## Figures and Tables

**Figure 1 microorganisms-11-00635-f001:**
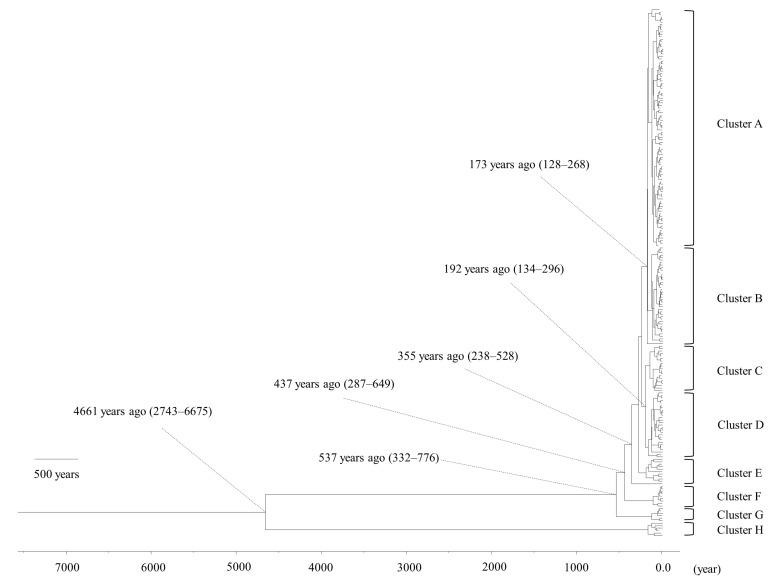
Phylogenetic tree of *bla*_PDC_, constructed using the Bayesian Markov Chain Monte Carlo method. The scale bar shows time (in years ago). The parentheses present the 95% highest posterior density (HPD).

**Figure 2 microorganisms-11-00635-f002:**
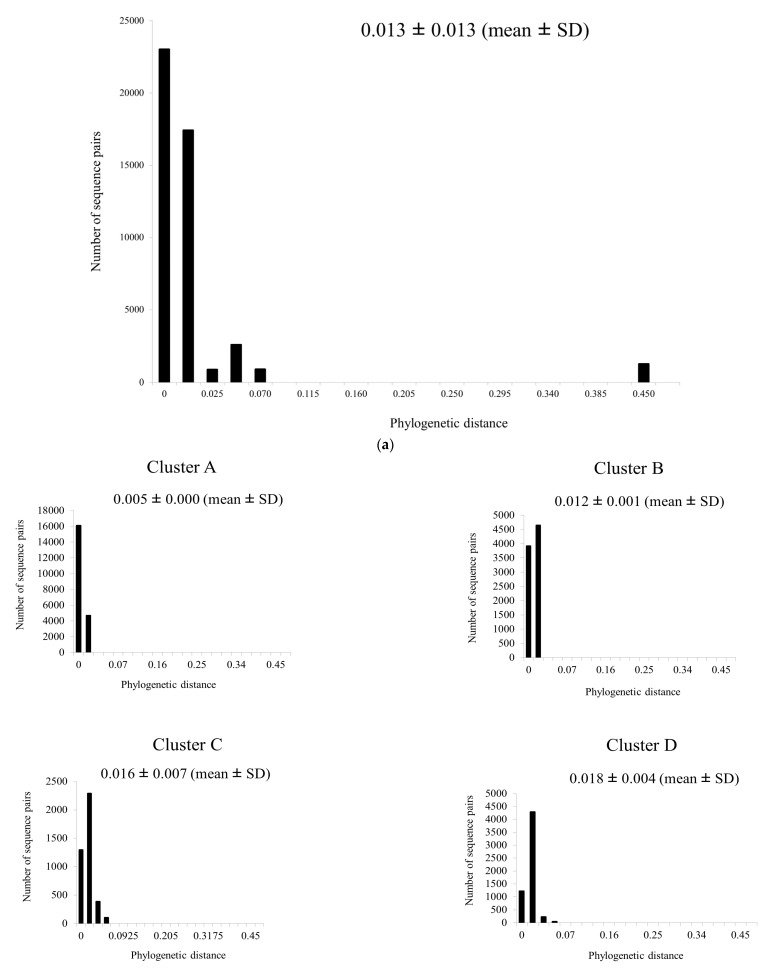
Phylogenetic distances for *bla*_PDC_ for (**a**) all samples and (**b**) each cluster. The y- and x-axes show the number of sequence pairs and phylogenetic distances, respectively.

**Figure 3 microorganisms-11-00635-f003:**
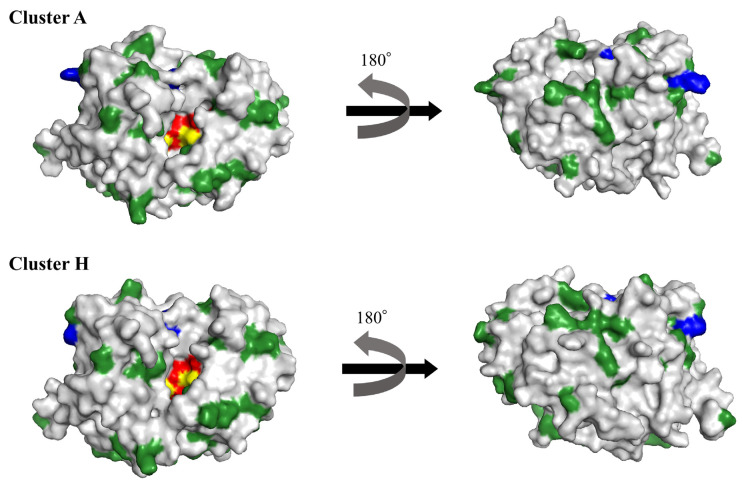
PDC structural models of the representative samples selected from clusters A and H. The active sites on the proteins are colored red. Positive and negative selection sites are shown in blue and dark green, respectively. The superimposed sites between active sites and negative selection sites are indicated in yellow.

**Figure 4 microorganisms-11-00635-f004:**
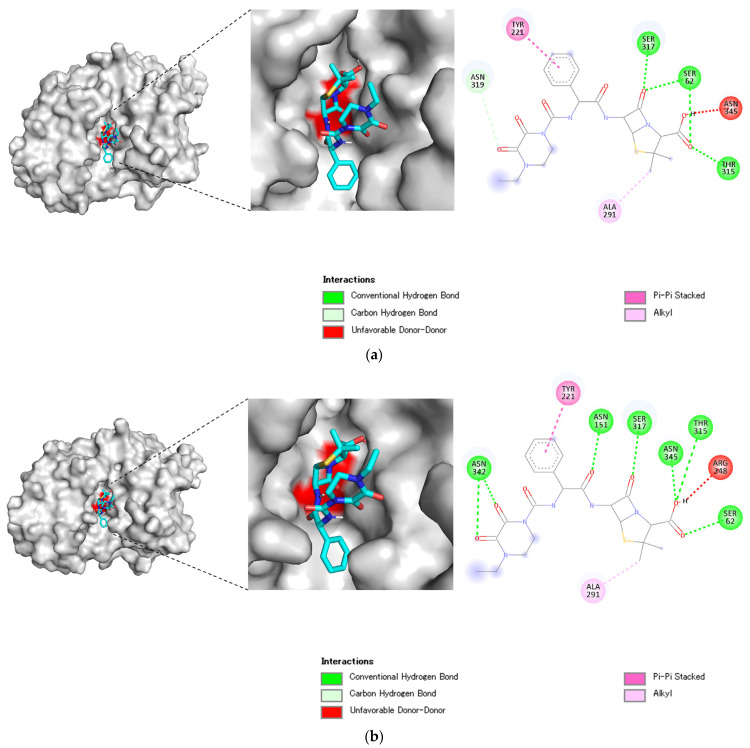
Structural model-based binding between piperacillin and representative PDC from clusters (**a**) A and (**b**) H. The three-dimensional (3D) structures of piperacillin are presented as stick models. The active sites on the protein surface are indicated in red.

## Data Availability

The data presented in this study are available on request from the corresponding author.

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
