# Peer review of "Molecular Evolutionary Analyses of the Pseudomonas-Derived Cephalosporinase Gene"

_microorganisms, 2023, doi:10.3390/microorganisms11030635_

Round 1

Reviewer 1 Report

The manuscript "Molecular evolutionary analyses of the Pseudomonas aeruginosa ampC gene" presents an analysis of the PDC class C beta-lactamase family from an evolutionary point of view.

This is a very interesting paper, complete, clear and well described. However, there are significant changes that are required to make it publishable (see below):

- line 26-27: This enzyme family has a specific name, PDC, and to date 533 allele numbers were assigned for this family, according to the BLDB (http://bldb.eu/BLDB.php?prot=C#PDC), CARD (https://card.mcmaster.ca/ontology/36237) or NCBI AMRDB (https://www.ncbi.nlm.nih.gov/bioproject/PRJNA313047). Please use this name (PDC) in the title, abstract, and throughout the manuscript.
- line 35: the binding affinity reported for docking calculations is not reliable. Please replace it with "the same binding mode", if this is what is seen in the docking results.
- lines 68-76: in GenBank, genomic data (e.g. for beta-lactamases) are deposited either as individual sequences for a given gene, or together with the whole genome. Here you are talking about strains, which means that you have considered only the PDC alleles deposited together with the whole genome. But many of them have been deposited only as individual sequences. Please include them too in the study, or clarify the text if necessary. Moreover, in many cases, a single PDC enzyme has been isolated at different times and places. If you considered only the first isolation date, this should be clearly stated in the text.
- line 99: you should specify that this is a Pseudomonas aeruginosa-specific protein, and you should use the PDC naming scheme, as mentioned above (this is PDC-1).
- line 107: structural models can be now generated also with AlphaFold, which is generally better than MODELLER in the conformation of side chains - this is important given that you have not optimized these models with molecular dynamics prior to docking studies. The pre-computed AlphaFold models for many proteins can be retrieved from the UniProt entry of the corresponding protein (see the Structure section) or from the AlphaFold database at EBI (https://alphafold.ebi.ac.uk/).
- lines 110-111: the binding site of class C beta-lactamases contain three main conserved motifs, and you mention only one (the SxxK motif). See the reference papers in the field (e.g. DOI 10.1128/cmr.00150-21, 10.1128/AAC.01841-19, 10.1128/CMR.00036-08, 10.1128/AAC.46.1.1-11.2002 - two of these papers are cited in the manuscript as references 29 and 30) and add the two other motifs.
- figure 1: specify the main PDC representative corresponding to each cluster A to H
- lines 175-176: you should include in the manuscript an alignment of selected sequences A to H, discuss if there are differences in the binding sites, and how these differences would affect the binding of substrate (then correlate with the docking results)
- lines 183, and 190: the analysis of interactions in the docking results is well described, but you should remove the sentences with the binding affinity, as this is not a reliable information.
- line 228: please remove the reference to the binding affinity
- line 241: references 42 and 43 are not pertinent for molecular dynamics simulations of beta-lactamases. There are many of them in the literature, I will give here only a few examples: DOI 10.1016/j.bpj.2012.09.009, 10.1021/bi051600j, 10.1021/acs.jpcb.1c05988, 10.7554/eLife.66567 ...
- ref 29: remove "Table of Contents"
- ref 39 is not complete. Please add the page (1660).

In conclusion, I would recommend the publication of this manuscript in Microorganisms, only after major revisions according to the comments presented above.

Author Response

Response to the Reviewer 1

  Thank you for reviewing our manuscript. According to your suggestions, we have revised the manuscript. Revisions are indicated as blue text. We hope that the revised manuscript is acceptable for publication in this journal.

- line 26-27: This enzyme family has a specific name, PDC, and to date 533 allele numbers were assigned for this family, according to the BLDB (http://bldb.eu/BLDB.php?prot=C#PDC), CARD (https://card.mcmaster.ca/ontology/36237) or NCBI AMRDB (https://www.ncbi.nlm.nih.gov/bioproject/PRJNA313047). Please use this name (PDC) in the title, abstract, and throughout the manuscript.

Reply 1-1

We thank the reviewer for pointing this out. According to your suggestion, we have replaced “AmpC-type β-lactamase” with “PDC”.

- line 35: the binding affinity reported for docking calculations is not reliable. Please replace it with "the same binding mode", if this is what is seen in the docking results.

Reply 1-2

We agree with your suggestion. Thus, we have replaced “the same binding affinity (−9.0 kcal/mol)” with “the same binding mode”.

- lines 68-76: in GenBank, genomic data (e.g. for beta-lactamases) are deposited either as individual sequences for a given gene, or together with the whole genome. Here you are talking about strains, which means that you have considered only the PDC alleles deposited together with the whole genome. But many of them have been deposited only as individual sequences. Please include them too in the study, or clarify the text if necessary. Moreover, in many cases, a single PDC enzyme has been isolated at different times and places. If you considered only the first isolation date, this should be clearly stated in the text.

Reply 1-3

Thank you for your comments. Our genome data includes the whole genome as well as the partial genome. In addition, we left the genome sample that was first isolated when the sequence was homologous. Accordingly, we have replaced “strain” with “sample” and added new sentences to the revised manuscript as follows:

Lines 74: the genome that was first isolated was also left.

- line 99: you should specify that this is a Pseudomonas aeruginosa-specific protein, and you should use the PDC naming scheme, as mentioned above (this is PDC-1).

Reply 1-4

We agree with your comments and have reflected the comments in the revised manuscript.

- line 107: structural models can be now generated also with AlphaFold, which is generally better than MODELLER in the conformation of side chains - this is important given that you have not optimized these models with molecular dynamics prior to docking studies. The pre-computed AlphaFold models for many proteins can be retrieved from the UniProt entry of the corresponding protein (see the Structure section) or from the AlphaFold database at EBI (https://alphafold.ebi.ac.uk/).

Reply 1-5

Thank you for your suggestions and for sharing the accessible database with us. We also think that AlphaFold is a powerful tool for predicting the 3D structure of a protein. However, the 3D structures of the protein selected from clusters A and H were not found in the UniProt and the AlphaFold database. Moreover, we could not perform AlphaFold2 because it requires high computing capability. In contrast, we could construct the structural models (representative PDC in clusters A and H) using its abridged version web server, ColabFold (https://colab.research.google.com/github/sokrypton/ColabFold/blob/main/AlphaFold2.ipynb). However, it is unclear whether these structural models are more reliable than homology modeling generated model. For this reason, we did not alter the structural models. Finally, your suggestion has been reflected in the discussion section as a limitation of the present study as follows:

Lines 243-250: This study has computing capability-related limitations that need to be addressed in the future. The first is the reliability of structural models. The homology modelling generates a good stereochemical protein model but limits the prediction of the native structure accurately from its sequence and template structure alone [43]. Quite recently, a new powerful tool of structural modelling using deep learning techniques, Alphafold2, has become available [44]. AlphaFold2 may exhibit a smaller structural gap between native and target proteins than homology modelling [44]. However, we could not perform AlphaFold2 because it requires high computing capability.

- lines 110-111: the binding site of class C beta-lactamases contain three main conserved motifs, and you mention only one (the SxxK motif). See the reference papers in the field (e.g. DOI 10.1128/cmr.00150-21, 10.1128/AAC.01841-19, 10.1128/CMR.00036-08, 10.1128/AAC.46.1.1-11.2002 - two of these papers are cited in the manuscript as references 29 and 30) and add the two other motifs.

Reply 1-6

Thank you for your comment. We have incorporated the information as follows:

Lines 113-115: Based on prior reports [29-32], three active sites of PDC were identified as follows: Ser62, Val63, Ser64, and Lys65 (here, motif 1); Tyr149, Ser150, and Asn151 (here, motif 2); Lys314, Thr315, and Gly316 (here, motif 3).

- figure 1: specify the main PDC representative corresponding to each cluster A to H

- lines 175-176: you should include in the manuscript an alignment of selected sequences A to H, discuss if there are differences in the binding sites, and how these differences would affect the binding of substrate (then correlate with the docking results)

Reply 1-7

We appreciate your comment. According to your suggestion, we have revised figure 1 and added new sentences in the revised manuscript (lines 90-96, 146-147, and 230-231)

- lines 183, and 190: the analysis of interactions in the docking results is well described, but you should remove the sentences with the binding affinity, as this is not a reliable information.

Reply 1-8

Thank you for your valuable comments. Reflecting your suggestion, we have revised the manuscript in abstract and discussion sections. Also, we have omitted the sentences regarding the binding affinity in result section and have mentioned results of docking simulation analysis only. However, to emphasize the calculated binding affinity and distinguish it from the native binding affinity, we have used the term “compute” or “computational”. The revised paragraph is as follows:

Lines 197-198: The computational binding affinity was estimated to be −9.0 kcal/mol.

Line 205: The binding affinity was computed as −9.0 kcal/mol.

- line 228: please remove the reference to the binding affinity

Reply 1-9

We agree with you and have revised our manuscript to reflect your suggestion.

- line 241: references 42 and 43 are not pertinent for molecular dynamics simulations of beta-lactamases. There are many of them in the literature, I will give here only a few examples: DOI 10.1016/j.bpj.2012.09.009, 10.1021/bi051600j, 10.1021/acs.jpcb.1c05988, 10.7554/eLife.66567 ...

Reply 1-10

We agree with you. As per your suggestion, we have changed references in the revised manuscript.

- ref 29: remove "Table of Contents"

- ref 39 is not complete. Please add the page (1660).

Reply 1-11

Thank you for pointing out this. We have incorporated the suggested changes.

Reviewer 2 Report

In the present study, Shirai et al. describe the first report deeply exploring the evolutionary analysis of ampC in P. aeruginosa. The article is well present, with proper methodologies and conclusions, and also brings relevant data related to one of the main resistance mechanisms used by several P. aeruginosa strains. I have only minor suggestion that may improve the manuscript presentation: 

Introduction: 

The 1st paragraph present an information overly repeated and well understood. Focus on adding references related to the article. 

Methods: 

Supplementary table including information on the 215 strains (1963 - 2021) should be provided. 

Discussion:

The 1st paragraph mostly repeats information from results section. Please rewrite. 

Please discuss in a more direct way how the present study have impacts on patients treatments and other healthcare related issues. 

Author Response

Response to the Reviewer 2

  Thank you for reviewing our manuscript. According to your suggestions, we have revised the manuscript. Revisions are indicated as blue text. We hope that the revised manuscript is acceptable for publication in this journal.

Introduction:

The 1st paragraph present an information overly repeated and well understood. Focus on adding references related to the article.

Reply 2-1

   Thank you for your valuable comment. According to your suggestion, we have revised introduction section.

Methods:

Supplementary table including information on the 215 strains (1963 - 2021) should be provided.

Reply 2-2

   We agree with your suggestion. Accordingly, we have included the genome sample information in supplemental Table 1 (lines 75-76).

Discussion:

The 1st paragraph mostly repeats information from results section. Please rewrite.

Reply 2-3

 Thank you for your recommendation. As per your suggestion, we have revised discussion section as follows:

Lines 240-242: Furthermore, deeper insight into the molecular mechanisms of PDC in evolutionarily selected variants will pave the way towards developing promising β-lactam/β-lactamase inhibitor against P. aeruginosa.

Lines 261-263: Although the methodology of the present study is subject to these limitations, our molecular evolution and docking simulation analyses will guide the designing of effective therapies targeting PDC.